# A multi-attribute method for ranking influential nodes in complex networks

**Adib Sheikhahmadi[1], Farshid Veisi[1], Amir Sheikhahmadi[1]***, Shahnaz Mohammadimajd[2]**

**1** Department of Computer Engineering, Sanandaj Branch, Islamic Azad University, Sanandaj, Iran,
**2** Department of Mathematics, Sanandaj Branch, Islamic Azad University, Sanandaj, Iran

\* asheikhahmadi@iausdj.ac.ir

## Abstract

Calculating the importance of influential nodes and ranking them based on their diffusion power is one of the open issues and critical research fields in complex networks. It is essential to identify an attribute that can compute and rank the diffusion power of nodes with high accuracy, despite the plurality of nodes and many relationships between them. Most methods presented only use one structural attribute to capture the influence of individuals, which is not entirely accurate in most networks. The reason is that network structures are disparate, and these methods will be inefficient by altering the network. A possible solution is to use more than one attribute to examine the characteristics aspect and address the issue mentioned. Therefore, this study presents a method for identifying and ranking node's ability to spread information. The purpose of this study is to present a multi-attribute decision making approach for determining diffusion power and classification of nodes, which uses several local and semi-local attributes. Local and semi-local attributes with linear time complexity are used, considering different aspects of the network nodes. Evaluations performed on datasets of real networks demonstrate that the proposed method performs satisfactorily in allocating distinct ranks to nodes; moreover, as the infection rate of nodes increases, the accuracy of the proposed method increases.

**Data Availability Statement:** All datasets files are available from the http://konect.cc/ database

**Funding:** The author(s) received no specific funding for this work.

## 1. Introduction

Many people use Social networks to communicate with friends, exchange opinions, and share information. The appealing environments of these networks have encouraged companies, political figures, and others to employ them for broadcasting innovations, advertising, and promoting their products [1]. Given people's tendency to have more trust in friends and acquaintances, many companies prefer to spread out their messages through individuals in a network [2]. Finding individuals who can maximize diffusion has always been of a great concern to these companies [3]. Such people are referred to as influential nodes. Finding influential nodes and utilizing them to indicate the advertisement process is a remarkably effective way of increasing the number of people who become aware of the advertised content [4]. Therefore, evaluating and ranking nodes' diffusion power in a network to propagate messages in online social networks have become a critical research topic in various sciences [5]. This

**Competing interests:** NO authors have competing interests The authors have declared that no competing interests exist.

problem comprises two sub-problems: 1- assessing the diffusion power of network nodes and ranking users based on it. 2- selecting an optimal subset of users to maximize the diffusion process [6]. The present study focuses on the first sub-problem. Thus far, nobody has presented a comprehensive and acceptable definition for influential nodes [7]. Some studies label high diffusion power as influential, while others label opinion leaders as people who can make others accept something by accepting it themselves as such [8]. This study uses the first definition, similar to many other studies Therefore, influential nodes are individuals who can propagate an advertisement message in the network with a high diffusion power.

There have been many methods to evaluate the diffusion power of network users that primarily use structural network information because they lack access to network information [9]. These methods consider nodes with a better place in the network as more influential [10]. However, the main problem of these methods is selecting the proper attribute to determine the diffusion power of nodes, considering the relatively high number of nodes and connections between them [11]. Many of these methods for assessing the diffusion power of nodes regard node from one aspect to calculate its influence based on an attribute [12]. These methods are only well-suited to some networks [13], and lose their effectiveness when the network changes.

These attributes could be local, semi-local, and global [14]. In the local attribute, the power of diffusion is calculated based on the neighbors of nodes. In contrast, global attributes measure the impact of the node using all nodes' information. The third class of attributes, known as a semi-local attribute, has been presented to reach a compromise between these two groups. This attribute takes into account information from multiple levels of a node's neighborhood to calculate diffusion power. For large-scale complex networks, global feature-based methods are unsuitable due to the high time complexity [15]. The local and semi-local methods are adequately faster, even though using only one local or semi-local cannot provide sufficient accuracy in dealing with various types of networks.

Ranking influential nodes can be considered as a Multi-Attribute Decision Making (MADM) problem in which the different attributes of each node can be used as influential criteria in decision-making. Thus, the primary hypothesis is that considering multiple local and semi-local features and treating them as a MADM problem can improve the performance of the method in comparison to methods that consider only one feature. This present study presents a method for determining and ranking the diffusion power of nodes that utilizes several different attributes. For comparing and ranking nodes according to their various dimensions, the proposed method uses the Elimination and Choice Translating Reality (ÉLECTRE) method, a family of MADM techniques. The ÉLECTRE method, also known as approximate dominance, is one of the MADM methods. It was first introduced by Benayoun in 1966 and then developed by researchers named Roy and Van Delf. This method evaluates all options by unranked comparisons, and the uninfluential ones are eliminated. All these steps are based on a coordinated set and an uncoordinated set, which gives the method its alternative title of coordination analysis. Concerning the time complexity of the ÉLECTRE method, the present study employs the simplified ÉLECTRE method improving computational efficiency and reducing time complexity while delivering the same performance as the ÉLECTRE method. The innovations introduced in this paper are as follows:

1. Identifying and extracting structural attributes from the network.

2. Ranking nodes based on different aspects of the network structure using several attributes.

3. Comparing and ranking network nodes using the simplified ÉLECTRE Multi-Attribute Decision Making method.

The related works will be reviewed first in the rest of this study then; section 3 introduces the proposed method and its components. In section 4, the proposed method will be evaluated, and a summary of the work will be presented in section 5.

## 2. Related works

Many methods have been proposed to measure the diffusion power of the nodes in a network. In most of these methods, the network structure and the strategic location of nodes have been used to determine their diffusion power. In these methods, the better position of the node, the more diffusion power in the following, some of these methods are mentioned.

In High Degree, which uses the degree of each node to calculate its centrality, it assumes nodes with a higher number of connections or friends are more influential [16]. In degree centrality, local information of nodes is used. In Closeness Centrality, which is a global method for identifying influential nodes in complex networks, the average distance between each node and all the other nodes in the graph is calculated. The less distance between a given node and others, the more influential it is. This method is highly time-consuming in large dynamic networks and has high computational complexity. Efforts have been made to improve the closeness centrality using the local structure of nodes, aiming to reduce its computational complexity. In [17], a new ranking method called Bridge Rank is proposed that calculates the local centrality of each node. Ref [18], first specifies all communities in the network and, by ignoring the relationships between communities, identifies a node as the local critical one according to the applied centrality metric. Next, by taking into account the edges between communities, a node is selected as the gateway, and the network nodes will be ranked based on the sum of the shortest distances from obtained critical nodes.

The K-Shell method claims that nodes in the center have a higher diffusion power [10]. Therefore, it allocates a number to each node based on its closeness to the center. Then, it uses these numbers to rank nodes and determines their diffusion power. In other words, nodes with higher numbers are stronger in this method. K-Shell ranks Nodes in the same Shell. It is assumed that the nodes in the higher Shell have higher diffusion power. The Mixed Degree Decomposition method (MDD) was proposed to improve K-Shell. In this method which is based on the K-Shell, the number of remaining edges $k_r$ and removed edges $k_s$ of each node are taken into account [19]. The corness method has also been proposed to improve the K-Shell method assumes nodes with more connections to neighbors located in the network center are much more powerful [11].

K-Shell IF method works based on the K-Shell method; however, separates nodes with the same $k_s$ by considering the iterations in each step of K-Shell; Then, it determines the diffusion power of each node by using the neighborhood concept up to one step [20]. In the Extended Weighted Degree Centrality method to determine the influence of nodes' diffusion, an extended weighted degree centrality method based on the degree of a node and its neighbors has been proposed [16]. In H-Index Centrality [21], The diffusion power of a graph node is calculated using a function based on its neighbor's degree. If y neighboring nodes have a degree greater than or equal to y, then the node y's H-index is considered. A metric is presented in the Extended H-index method [22], which uses the neighbors' information to determine the centrality of nodes through an expansion of the H-index concept. Sheikhahmadi et al. [5] Proposed the Mixed Core, Degree, and Entropy (MCDE) method. In this multi-attribute method, the diffusion power of neighbors is measured based on a combination of features including core number, degree, and level of Dispersion. Entropy-based Ranking Measure (ERM) is a semi-local method based on the hypothesis that nodes with high diffusion power have neighbors with high degrees; additionally, the neighbors of these nodes possess a degree

of monotonicity. ERM calculates the degree entropy of one- and two-step neighbors of a node. Then the centrality of each node is calculated based on these two criteria [22].

Due to the lack of information provided by the K-Shell attribute about the topological positions of nodes in the graph, an index called Hierarchical K-Shell (HKS) [23] has been proposed. This method aims to determine a nodes' topological position by extracting structural information ignored by K-Shell, then estimating the diffusion power of each node using that information and the nodes are ranked.

Namtirtha et al. [24] proposed the K-Shell degree neighborhood method by assigning weights to graph edges using node degree and the K-Shell index of the nodes at the ends of each edge. Then, to measure the influence power of each node, they calculated the sum of the weights of all edges connected to that node. Maji [25], In a similar work to [24], However, instead of adjusting parameters, used a measure based on the network's average degree and K-Shell and a combination of a K-Shell index and degree of nodes to weigh the graph edges.

The gravity formula states that the force which two objects exert on each other is directly related to their mass and inversely related to their distance. Based on this fact, Ma et al. [26] observed a nodes' effect on spreading activity. In order to propose a gravity measurement formula, the K-Shell value of a node was used as the mass and the shortest path distance between each pair of nodes as the distance.

Li et al. [27] proposed a gravity centrality (GC) model based on the gravity formula, which assumes a node degree as its mass and its shortest path distance as the distance between each pair of nodes. With gravitational centrality, nodes are only interactive based on their degrees and distances, indicating they have the same gravity. Each node may have a different absorption capacity in the real world. Liu et al. [28] improved this model by considering the weight of each node in the network and identified a new centrality measure called WGC that is more relevant to real-world networks.

Yang et al. [29] also took the location of nodes into consideration, it means a node in the center of the network's center is more likely to attract other nodes than a node on the periphery. Therefore, they proposed an improved gravity model; based on the K-shell algorithm to identify influential nodes in networks. The differences in location between nodes, modeled by differences in K-shell values, are used as attraction coefficients, which adjust the attractiveness of central nodes in the networks. The proposed approach combines Local and global information.

MADM methods can be used to evaluate the diffusion power of network nodes based on a variety of dimensions. Du et al. [30] used the Technique for Order Preference By Similarity to Ideal Solution(TOPSIS) method to identify influential nodes in complex networks. They chose nodes with the least distance from the optimal solution and the most distance from the worst solution simultaneously. Liu et al. [31] utilized a combination of relative entropy and TOPSIS to evaluate the diffusion power of nodes and applied their method to several real-world complex networks. Yang et al. [32] employed gray correlation analysis to determine the weights of evaluation indices and presented a dynamic weighted TOPSIS algorithm for finding nodes with high diffusion power in complex networks. Yang et al. [33] presented an integrated measurement method for identifying influential nodes in a complex network by combining the entropy weighting method with the Vlse Kriterijumska Optimizacija Kompromisno Resenje (VIKOR) method, which means multi-criteria optimization and compromise solution, in Serbian.

## 3. The proposed method

Fig 1 depicts the general procedure of the proposed method. The proposed method extracts important structural attributes that identify nodes from the input social network. As extracting and using all the attributes to compare and rank the influential nodes is time-consuming, a subset of more accurate features is selected. In the next step, the ÉLECTRE method is used for

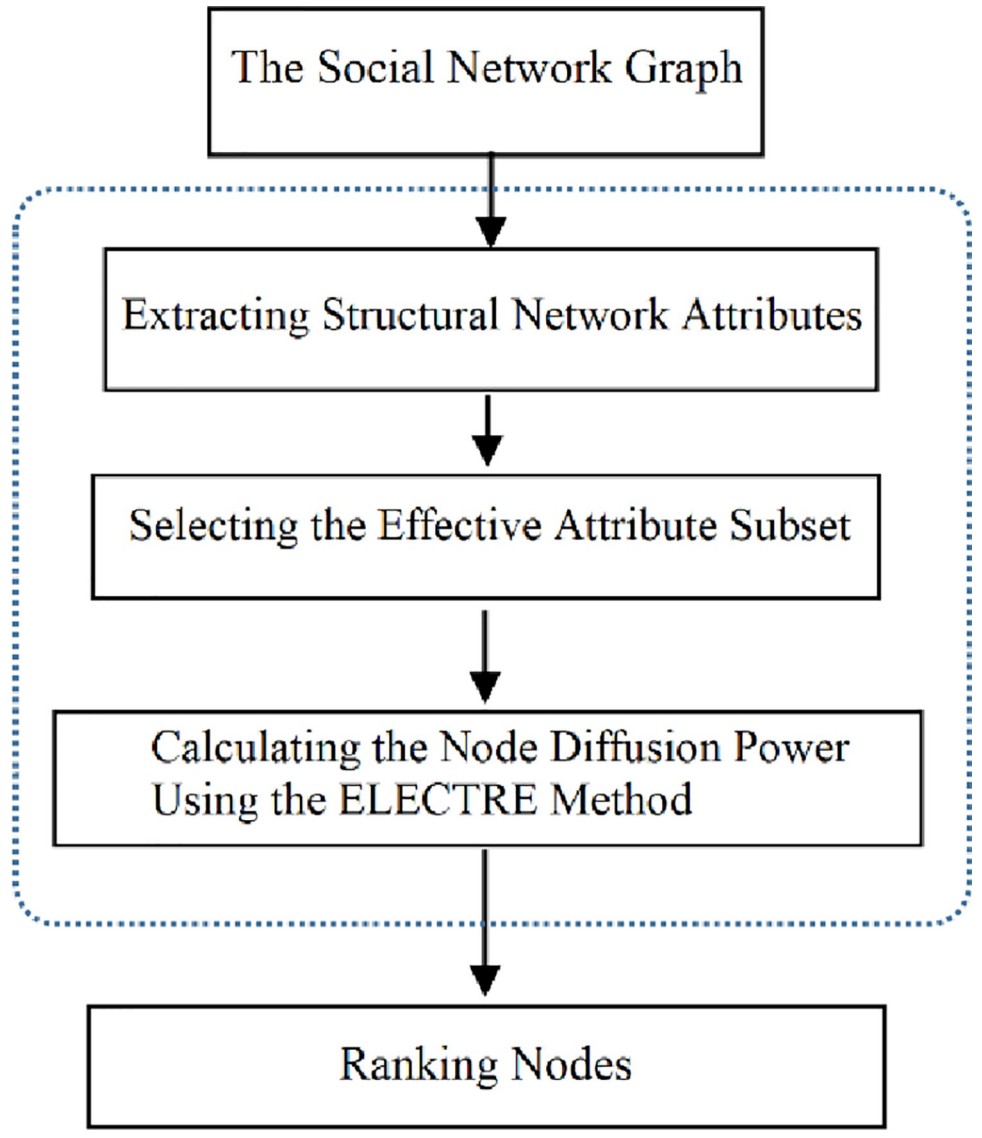

**Fig 1. General procedure of identifying and ranking influential nodes.**

comparing and ranking node scores. The following section will examine each part of the proposed method in detail.

### 3.1. Input network

The input network is a two-column file where the first column contains the source node's number and the second column contains the destination node's number. For example, Table 1 shows part of the input used in the method.

For example, there is a link between nodes 1 and 2, as shown in the first row of Table 1.

### 3.2. Extracting the structural network attributes

There are several methods for calculating the diffusion power of nodes based on the network structure and the position of each node. Many of these methods are single-attribute methods.

**Table 1. A sample from the input data.**

| Node To | Node To |
| --- | --- |
| 1 | 2 |
| 1 | 3 |
| 1 | 4 |
| 2 | 3 |
| 2 | 4 |
| 3 | 4 |

In other words, these methods calculate diffusion power for nodes in the network by only using one attribute. As pointed out earlier, these methods are only effective in some networks and will not work if the network changes. In this section, several methods with sufficient accuracy and acceptable execution time have been selected. The methods utilized in this section are as follows: degree [34], K-Shell ($k_s$) [10], Coreness [11], MDD [19], K-Shell IF [20], H-index [35], HKS [23], ERM [9], and Gravity [27]. It should be noted that due to many available methods, this section only considered local or semi-local methods whose reported time for calculation is acceptable.

### 3.3. Selecting the effective attribute subset in node diffusion evaluation

A number of effective features are selected based on the diversity of extracted features in this part to be used in the next step. To provide better understanding, data belonging to the Zachary karate club is shown in the graph in Fig 2.

In the following, the structural features discussed in section 3.2 will be calculated for this graph, and a method for selecting the most effective subset. The obtained values of the other calculated characteristics for each node are shown in Table 2. Apart from the values obtained for each attribute, the diffusion power of each node is also calculated and displayed in the last column of Table 2. To evaluate the spreading power of a node, either the network must be monitored in real-time, or diffusion models must be employed. Since a network cannot be monitored except by network owners in most cases, researchers tend to use epidemic models to measure the diffusion power of nodes. Throughout this section, the susceptible-infected-

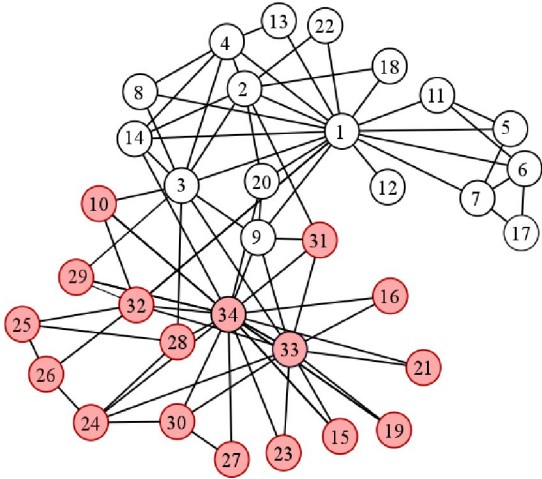

**Fig 2. The Zackary karate club.**

**Table 2. Obtained values for other structural characteristics.**

| node | Degree | kshell | coreness | MDD | kshell_if | H-index | HKS | ERM | Gravity | Spread Power (SIR) |
|------|--------|--------|----------|------|-----------|---------|------|---------|---------|--------------------|
| 1 | 16 | 4 | 250 | 11.2 | 534.5 | 5 | 7441 | 326.571 | 196 | 4.86 |
| 2 | 9 | 4 | 187 | 6.9 | 413.5 | 4 | 5021 | 235.639 | 124 | 4.03 |
| 3 | 10 | 4 | 226 | 7.6 | 505.5 | 5 | 5623 | 278.354 | 144 | 5.12 |
| 4 | 6 | 4 | 160 | 5.1 | 370.667 | 4 | 3728 | 183.816 | 88 | 3.89 |
| 5 | 3 | 3 | 71 | 3 | 173 | 3 | 1340 | 68.4286 | 30 | 2.46 |
| 6 | 4 | 3 | 77 | 3.4 | 185.5 | 3 | 1412 | 72.6049 | 36 | 2.87 |
| 7 | 4 | 3 | 77 | 3.4 | 185.5 | 3 | 1412 | 72.6049 | 36 | 2.94 |
| 8 | 4 | 4 | 138 | 4 | 329.333 | 4 | 2766 | 145.947 | 64 | 3 |
| 9 | 5 | 4 | 184 | 4.7 | 417.333 | 4 | 3243 | 190.896 | 80 | 4.17 |
| 10 | 2 | 2 | 84 | 2 | 178.667 | 2 | 1404 | 85.0739 | 16 | 2.24 |
| 11 | 3 | 3 | 71 | 3 | 173 | 3 | 1340 | 68.4286 | 30 | 2.3 |
| 12 | 1 | 1 | 49 | 1 | 130 | 1 | 752 | 44.5235 | 4 | 1.62 |
| 13 | 2 | 2 | 71 | 2 | 176 | 2 | 1344 | 71.9343 | 16 | 2.28 |
| 14 | 5 | 4 | 186 | 5 | 432 | 5 | 3387 | 190.304 | 80 | 3.96 |
| 15 | 2 | 2 | 83 | 2 | 162.667 | 2 | 1238 | 82.4038 | 16 | 2.4 |
| 16 | 2 | 2 | 83 | 2 | 162.667 | 2 | 1238 | 82.4038 | 16 | 2.86 |
| 17 | 2 | 2 | 24 | 2 | 44 | 2 | 602 | 25.0211 | 12 | 2.04 |
| 18 | 2 | 2 | 80 | 2 | 196 | 2 | 1391 | 77.8193 | 16 | 2.34 |
| 19 | 2 | 2 | 83 | 2 | 162.667 | 2 | 1238 | 82.4038 | 16 | 2.72 |
| 20 | 3 | 3 | 128 | 3 | 292.167 | 3 | 2012 | 122.177 | 36 | 2.84 |
| 21 | 2 | 2 | 83 | 2 | 162.667 | 2 | 1238 | 82.4038 | 16 | 2.19 |
| 22 | 2 | 2 | 80 | 2 | 196 | 2 | 1391 | 77.8193 | 16 | 2.77 |
| 23 | 2 | 2 | 83 | 2 | 162.667 | 2 | 1238 | 82.4038 | 16 | 2.29 |
| 24 | 5 | 3 | 119 | 4.1 | 240.167 | 4 | 2105 | 128.514 | 51 | 2.67 |
| 25 | 3 | 3 | 44 | 3 | 87 | 3 | 1109 | 55.3571 | 27 | 2.24 |
| 26 | 3 | 3 | 47 | 3 | 93 | 3 | 1123 | 56.9549 | 27 | 2.31 |
| 27 | 2 | 2 | 61 | 2 | 116.667 | 2 | 980 | 62.5179 | 14 | 1.92 |
| 28 | 4 | 3 | 110 | 3.7 | 238.167 | 3 | 1926 | 114.621 | 42 | 3.73 |
| 29 | 3 | 3 | 105 | 3 | 220.167 | 3 | 2005 | 112.481 | 33 | 2.7 |
| 30 | 4 | 3 | 107 | 3.7 | 210.667 | 3 | 1838 | 113.732 | 39 | 3.52 |
| 31 | 4 | 4 | 134 | 4 | 269.333 | 4 | 2583 | 146.587 | 64 | 3.72 |
| 32 | 6 | 3 | 161 | 5.1 | 359.167 | 3 | 2613 | 162.401 | 63 | 4 |
| 33 | 12 | 4 | 211 | 8.7 | 413.333 | 5 | 5448 | 275.448 | 140 | 4.98 |
| 34 | 17 | 4 | 234 | 11.9 | 433.667 | 5 | 7138 | 340.369 | 192 | 5.44 |

recovered (SIR) diffusion model is used. This model identifies the diffusion power of nodes by repeating the spreading process many times for each node, likely to be in keeping with reality.

In the next step, to determine the diffusion power, the correlation level between the list ranked by each feature and the list ranked by the SIR diffusion model is utilized to select the effective subset of indices. A higher correlation between these two lists indicates a more accurate attribute for determining node diffusion power. Here, Kendall's tau correlation coefficient is applied to see whether two ranking lists are correlated. Suppose $(x_1, y_1), (x_2, y_2), \ldots (x_n, y_n)$ are a set of pairs of ranks in two separate ranking lists, X and Y. For each pair $(x_i, y_i)$ and $(x_j, y_j)$ if $(x_i > x_j)$ and $(y_i > y_j)$ or $(x_i < x_j)$ and $(y_i < y_j)$ as concordant and If $(x_i > x_j)$ and $(y_i < y_j)$ or $(x_i < x_j)$ and $(y_i > y_j)$ are considered as discordant. Then the Kendall Tau value [36, 37] of two ranking lists, X and Y, is calculated using the relation $\tau(X, Y) = \frac{n_c - n_d}{1/2} (n)(n - 1)$ which $n_c$ and $n_d$ are

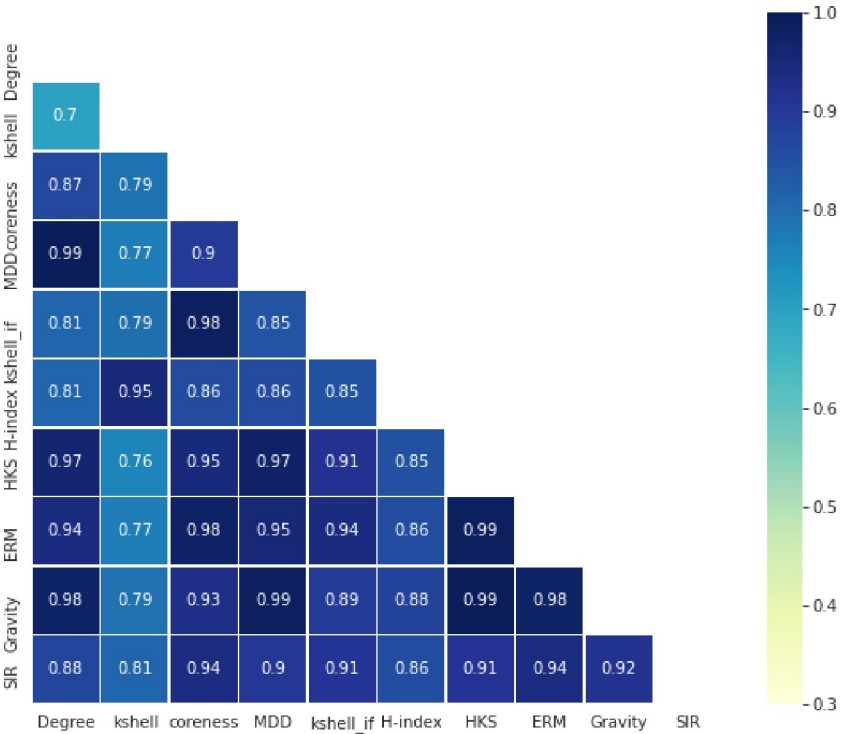

**Fig 3. The degree of correlation between the list ranked by each feature and the list ranked by the SIR diffusion model.**

the number of positive and negative pairs in the two ranking lists, respectively, and n is the size of the ranking vector.

The degree of correlation between the attributes extracted from Table 2 is presented in Fig 3.

Values in Fig 3 demonstrate that HKS, k-shell IF, Coreness, Gravity, and ERM are more accurate at ranking nodes than other features. Therefore, they can be considered effective subset features. The high correlation between the list ranked by these measures and real-world spreading is among the reasons for this selection. As an additional guarantee supporting this selection of features, Fig 4 illustrates the degree of correlation between each measure and the SIR model calculated for some of the datasets in Table 3.

Fig 4 demonstrates that HKS, k-shell IF, Coreness, Gravity, and ERM structural measures produce more accurate node rankings than others.

## 3.4. Calculating the node diffusion power using the ÉLECTRE method

AS Previously, five structural indices were selected from nine features as effective sets of features: HKS, k-shell IF, Coreness, Gravity, and ERM. the simplified ÉLECTRE method will be used to rank network nodes based on these attributes. The ÉLECTRE method or approximate dominance is a multi-criteria decision-making method.

The most significant advantage of the ÉLECTRE method over other decision support techniques is that it can be used to examine options for ordinal and more or less descriptive data. This method demonstrates the degree of dominance of one option over the others and is capable of utilizing incomplete data.

This method is implemented through the following steps:

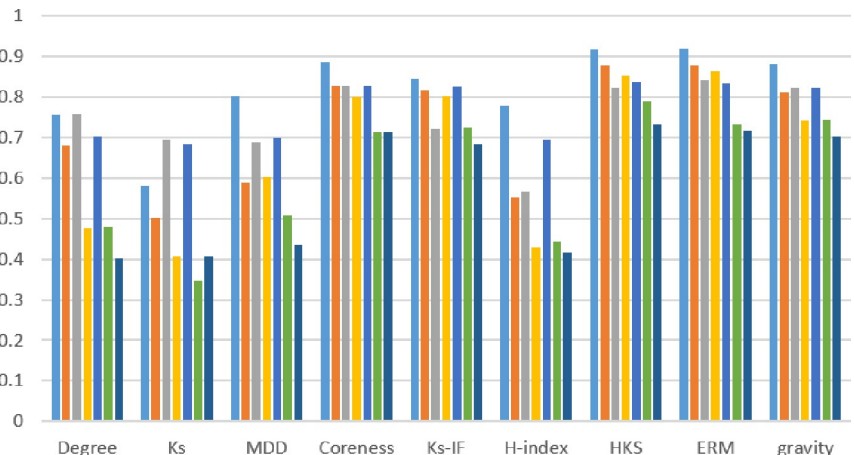

**Fig 4. Degree of correlation between each attribute and the SIR diffusion model.**

**Step One—Creating the Decision Matrix.** The decision matrix is created.

The number of nodes in the graph represents the number of rows, and the number of indices extracted from the network is the number of columns. Therefore, the decision matrix is created according to Eq 1.

$$X = \begin{bmatrix} x_{11} & \dots & x_{1n} \\ \dots & \dots & \dots \\ x_{m1} & \dots & x_{mn} \end{bmatrix} \tag{1}$$

Where $x_{ij}$ is the value of the $j$-th index for the $i$-th node.

**Step Two–Normalizing the Decision Matrix.** Due to the differences in dimensions between various centrality indices, the values for different measures will be normalized in this step. Normalization is done according to Eq 2:

$$r_{ij} = \frac{x_{ij}}{\sqrt{\sum_{i=1}^{m} x_{ij}^2}} \tag{2}$$

**Table 3. Applied datasets.**

| Network name | \|V\| (number of nodes) | \|E\| (number of edges) | highest network node degree | average node degree | Assortativity |
|---|---|---|---|---|---|
| Zebra | 27 | 111 | 14 | 8.2222 | 0.71770 |
| Karate | 34 | 78 | 17 | 4.5882 | -0.47561 |
| Contiguous | 49 | 107 | 8 | 4.3673 | 0.23340 |
| Dolphins | 62 | 159 | 12 | 5.1290 | -0.043594 |
| Copperfield | 112 | 425 | 49 | 7.5893 | -0.12935 |
| Netsciense | 379 | 914 | 34 | 4.8232 | -0.0817 |
| Elegans | 453 | 4,596 | 639 | 20.291 | -0.22582 |
| Euroroad | 1,174 | 1,417 | 10 | 2.4140 | 0.12668 |
| Chicago | 1,467 | 1,298 | 12 | 1.7696 | -0.50492 |
| Hamsterster | 2,426 | 16,631 | 273 | 13.711 | 0.047404 |
| PowerGrid | 4,941 | 6,594 | 19 | 2.6691 | 0.0034570 |
| PGP | 10,680 | 24,316 | 205 | 4.5536 | 0.23821 |

**Step Three—Determining the criteria Weight Matrix.** This step determines the attribute importance coefficient vector of criteria. Different methods, such as AHP and Shannon Entropy, can determine the attribute weights. In this study, Shannon's entropy method has been employed.

**Step Four—Determining the Normalized Weighted Decision Matrix.** The weighted decision matrix is obtained by multiplying the scale-free decision matrix with the criteria weights.

$$v_{ij} = w_j * r_{ij} \quad j = 1, 2, \ldots, n; \ i = 1, 2, \ldots, m$$

*Step Five*—**Forming a set of concordant and discordant criteria.** The attribute sets are divided into concordant and discordant subsets for each pair of nodes, k and e. The concordant set ($S_{ke}$) is a set of attributes that prefer node k to node e with the discordant set ($D_{ke}$) as its complementary set. The concordant set for positive and negative measures, respectively, is given by Eq 3.

$$S_{ke} = \{j | v_{kj} \geq v_{ej}\}$$
$$S_{ke} = \{j | v_{kj} \leq v_{ej}\}$$
(3)

The discordant set for positive and negative attributes is defined by Eq 4.

$$D_{ke} = \{j | v_{kj} < v_{ej}\} = J - S_{ke}$$
$$D_{ke} = \{j | v_{kj} > v_{ej}\} = J - S_{ke}$$
(4)

**Step Six—Creating the Concordant Matrix.** The concordant matrix is a square matrix as large as the number of options or graph nodes. Each element in this matrix is the concordant attribute between two nodes. The value of this attribute is the sum of the weights of the criteria in the concordant set. In other words, calculating the $C_{ke}$ concordant attribute requires a comparison between the k and e nodes and adding the attribute weights where k is preferred to e. In mathematical terms, the concordant attribute is calculated using Eq 5.

$$C_{ke} = \sum_{j \in S_{ke}} W_j$$
(5)

The concordant attribute indicates the superiority of node k over node e, and its value ranges from zero to one.

**Step Seven—Determining the Discordant Matrix.** The discordant matrix is a square matrix whose dimension is the number of nodes in the graph. Each element in this matrix is referred to as the discordant index between the two nodes. The value of this index can be calculated using Eq 6.

$$d_{ke} = \frac{\max_{j \in D_k} |v_{kj} - v_{ej}|}{\max_{j \in J} |v_{kj} - v_{ej}|}$$
(6)

**Step Eight—Creating the Concordant Dominance Matrix.** Step six depicted how to calculate the concordant attribute ($C_{ke}$). Now, this stage will determine a value for the concordant attribute known as the concordant threshold shown with $\bar{c}$. This concordant threshold is obtained by averaging all concordant attributes (the concordant matrix elements). In mathematical terms, the concordant threshold is calculated according to Eq 7.

$$\bar{C} = \sum_{k=1}^{m} \sum_{e=1}^{m} \frac{c_{ke}}{m(m-1)}$$
(7)

The concordant dominance matrix (F) is created based on the value of concordant threshold. If $C_{ke}$ is larger than $\bar{c}$, the superiority of node k over node e is acceptable.; Otherwise, node e has no superiority over e node. Therefore, the concordant dominance matrix elements are determined according to Eq 8.

$$f_{ke} = \begin{cases} 1 & c_{ke} \geq \bar{C} \\ 0 & c_{ke} < \bar{C} \end{cases} \tag{8}$$

**Step Nine—Creating The Discordant Dominance Matrix.**  The discordant dominance matrix (G) is created similarly to the concordant dominance matrix. Therefore, it must start by calculating the discordant threshold ($\bar{d}$) by averaging all discordant attributes (discordant matrix elements). In mathematical terms, the discordant threshold value is calculated using Eq 9.

$$\bar{d} = \sum_{k=1}^{m} \sum_{e=1}^{m} \frac{d_{ke}}{m(m-1)} \tag{9}$$

As stated in step seven, lower discordant attribute values $d_{ke}$ are better because discordant determines the superiority of node k over node e. If $d_{ke}$ is larger than $\bar{d}$, then the discordant value is too high, and it cannot be ignored. Therefore, the elements in the discordance domination matrix G are given by Eq 10.

$$g_{ke} = \begin{cases} 1 & d_{ke} \geq \bar{d} \\ 0 & d_{ke} < \bar{d} \end{cases} \tag{10}$$

Each member of matrix G determines the dominance relationship between nodes.

**Step Ten—Creating the Final Dominance Matrix.**  The final dominance matrix (H) is obtained according to Eq 11 by multiplying each element in the concordant dominance matrix (F) with the discordant dominance matrix (G).

$$h_{ke} = f_{ke} \cdot g_{ke} \tag{11}$$

**Step Eleven—Selecting the Best Option.**  The final dominance matrix (H) expresses the partial preferences of nodes. For instance, if $h_{ke}$ is one, in this case, the superiority of node k over node e is acceptable in both concordant and discordant states (superiority is larger than the concordant threshold and inferiority, or lack of concordant, is also less than the discordant threshold). However, node k still has a chance to dominate through other Nodes. The options can be ranked according to which node is more defeated over the other, dominates. Consequently, the sum of the rows of the H matrix represents the dominance of a node, whereas the sum of the columns represents the defeats of a node, which is derived from these two rank values assigned to each node. A positive number indicates more dominant nodes than defeated ones, while a negative number means the defeated nodes are more.

## 4. Evaluation

In order to evaluate the proposed method in this paper, the other compared methods have been implemented in Python 3.8 language programming and run on a system with a core i7 2.3 GHz processor and 16 GB of memory. For this evaluation,12 real-world datasets used, with

their characteristics listed in Table 3. The features for each dataset presented in Table 3 are, from left to right, as follows: the network name, the number of nodes, the number of edges, the highest network node degree, the average degree, and assortativity [26].

## 4.1. Evaluation criteria

The proposed method in this paper has been compared with other methods based on criteria used in other papers. The following criteria:

- Comparing the Node Diffusion Power obtained Using Different Methods with Their Real Diffusion power: This study uses the SIR diffusion model [38, 39] to calculate the real node diffusion power. The reason behind choosing this model is its widespread application in papers proposed in recent years [40]. This model simulates the message diffusion process in the real world and determines the real diffusion power of each node with many iterations for each node. Then, to evaluate the veracity and accuracy of the proposed algorithms, the ranking list proposed by the algorithm is compared with the ranking list calculated with the help of diffusion models. A high correlation between these two lists depicts the high algorithm accuracy in determining the node diffusion power and ranking them. This study uses Kendall's Tau [41] correlation coefficient to analyze the proposed algorithm's accuracy and correlation with the real ranking list. Given that the top-ranking nodes are more important than the low-ranking ones in these lists, a portion of the tests is reserved for examining the veracity of higher ranks in the list for this purpose, the similarity between the top c elements of list R ranked by each method and the top c elements in the real ranking list σ is calculated. The Jaccard similarity coefficient [42] is used in this section. This coefficient for the first c elements in lists X and Y is calculated using Eq (12).

$$J_c(X, Y) = \frac{|X(c) \cap Y(c)|}{|X(c) \cup Y(c)|} \tag{12}$$

X(c) is the set of elements in the list X at its initial rank.

- Allocating Distinct Ranks to Nodes with Different Diffusion Effects: according to this criterion, a method is better if it assigns fewer nodes in each rank. To assess the resolution of ranking, the monotonicity parameter (M) has been employed, which is defined according to Eq 13

$$M(R) = \left(1 - \frac{\sum_{r \in R} n_r * (n_r - 1)}{n * (n - 1)}\right)^2 \tag{13}$$

Where, N is the number of distinct ranks in list Rand $n_r$ is the number of nodes with a similar r rank in the list. The value of M will be zero if all nodes have the same rank, and M will be one if all nodes have a distinct rank. Also, to examine the performance of the proposed algorithms, each algorithm is executed 100 times on different networks, and their average execution time is compared with the other methods.

## 4.2. Test results

The results obtained from the tests conducted on the proposed method as compared with other methods. The methods are first compared by the accuracy of each method in ranking and then based on the resolution of node ranking.

**4.2.1. Method accuracy in ranking nodes.** To determine the accuracy of the methods, the ranking list produced by each method is compared with the ranking of influential nodes

**Table 4. The correlation coefficient between the ranked lists using each method and the ranked list using the SIR model.**

| Dataset | β | τ(ks,σ) | τ(MMD,σ) | τ(C_nc+,σ) | τ(ks-IF,σ) | τ(EW,σ) | τ(MCDE,σ) | τ(Electre,σ) |
|---|---|---|---|---|---|---|---|---|
| Zebra | 0.10 | 0.5670 | 0.6211 | **0.8462** | 0.8291 | 0.8348 | 0.8366 | **0.8462** |
| Karate | 0.15 | 0.5721 | 0.7112 | **0.8627** | 0.7772 | 0.7576 | 0.7976 | **0.8627** |
| Contiguous | 0.25 | 0.4048 | 0.7577 | 0.8971 | 0.9039 | **0.9320** | **0.9320** | **0.9320** |
| Dolphins | 0.15 | 0.5791 | 0.8154 | 0.9027 | 0.8636 | 0.9281 | 0.9381 | **0.9418** |
| Copperfield | 0.10 | 0.726 | 0.8399 | 0.9004 | 0.8652 | 0.9134 | 0.9155 | **0.9244** |
| Elegans | 0.01 | 0.6946 | 0.6886 | 0.8265 | 0.7216 | 0.8244 | **0.8344** | 0.8265 |
| Netsciense | 0.15 | 0.5018 | 0.5886 | 0.8263 | 0.8171 | **0.8938** | 0.8838 | 0.8884 |
| Email | 0.10 | 0.8126 | 0.8340 | **0.9305** | 0.8900 | 0.9162 | 0.9262 | **0.9305** |
| Euroroad | 0.35 | 0.4082 | 0.6015 | 0.8003 | 0.8024 | 0.8283 | 0.8383 | **0.8483** |
| Hamsterster | 0.03 | 0.6836 | 0.7007 | 0.8266 | 0.8245 | 0.8299 | 0.8361 | **0.8431** |
| PowerGrid | 0.30 | 0.3458 | 0.5255 | 0.7596 | 0.7727 | 0.7832 | 0.7812 | **0.7932** |
| PGP | 0.10 | 0.4073 | 0.4361 | 0.7144 | 0.6821 | 0.7220 | 0.7300 | **0.7357** |

obtained from the SIR model. The SIR model determines the real diffusion power of all nodes with many iterations, and based on that, the ranking list σ will be obtained. Given the stochastic nature of the process and in order to bring the results closer to reality, the SIR model is repeated $10^3$ times for each node $v_i$ in the graph, and the average number of improved nodes will be taken as the diffusion power of node $v_i$.

The Kendall tau correlation coefficient has been employed to determine the degree of correlation between the ranking list obtained from each method and the ranking list σ [43]. Table 4 depicts the Kendall-Tau correlation coefficient values between ranked nodes using each method, and the SIR ranked list. Each row in this table depicts the values for each network. Notably, higher vales determine a bigger similarity between obtained raking and reality.

The results from Table 4 show that the proposed method has a higher ranking accuracy than others in most datasets except the Netscience and Elegans, where it still had a performance close to the top method. Considering that different networks have diverse structural attributes and a single attribute performs well just in some network, using diverse structural attributes in the proposed method, which remarkably increases of the networks' accuracy. In other words, changing the network structure, unlike other methods, have no significant effect on the accuracy obtained by the proposed method.

The infection rate is an effective parameter in the SIR mode; therefore, the following section analyzes the β (infection rate) parameter effect on the proposed method's accuracy, and the results are presented in Fig 5. Considering numerous applied datasets, variations of this parameter are only analyzed on the Dolphins, Netscience, and PowerGrid datasets.

By increasing β, the infection rate of nodes will be increased, even though the spreading process will influence nodes in farther proximity. Furthermore, this method has a higher correlation than others because it consists of multiple attributes with the ÉLECTRE method to determine the node diffusion power; therefore, it will still have a higher correlation than others by increasing β and exerting changes in Networks. In the next test, the validity of the top c ranks of the ranking lists obtained from different methods is examined using the Jaccard similarity coefficient. The results of this test on the three networks of Netscience, Elegans, and PowerGrid are illustrated in Fig 6. In this test as well, the similarity coefficient of the top c ranks of the ranking list σ and the lists presented by various methods are examined by altering c. A shown in Fig 6 that the proposed methods have a higher validity and accuracy in the top ranks compared to other similar methods.

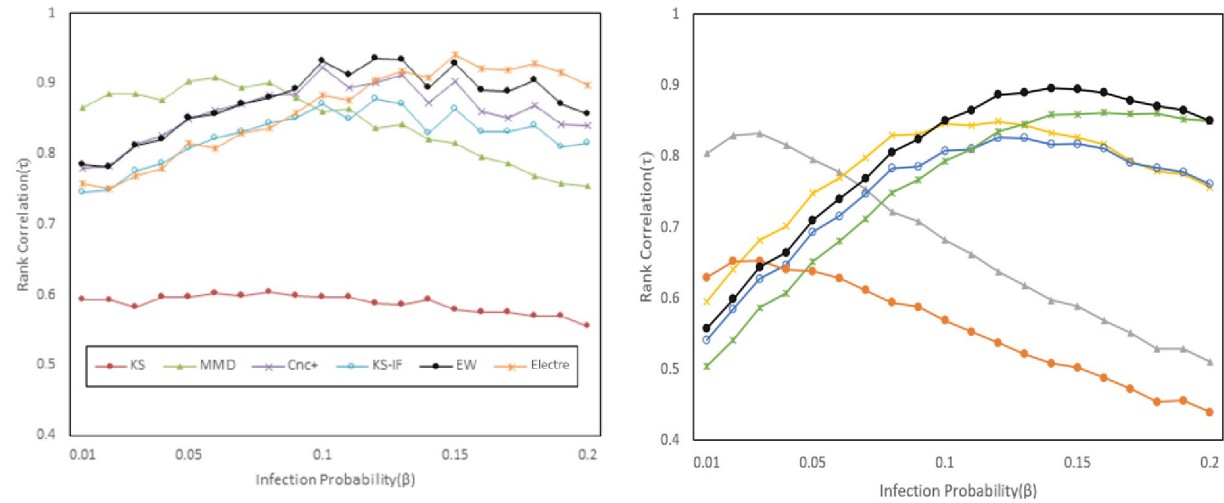

A. Dolphins network          B. Netscience network

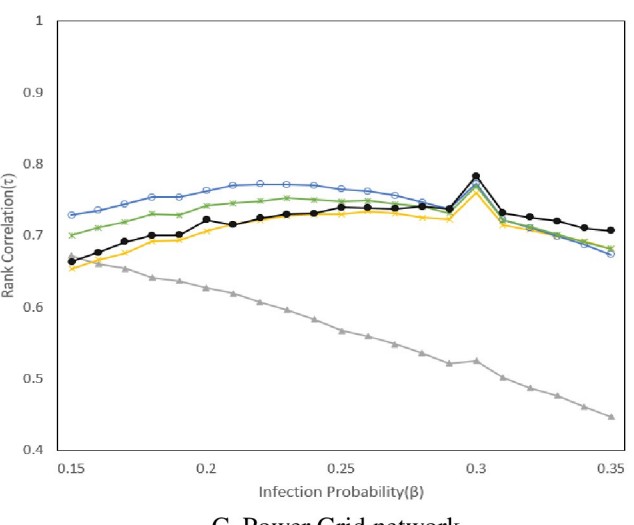

C. Power Grid network

**Fig 5. Parameter change effects on the proposed method's accuracy.** A. Dolphins network; B. Netscience network; C. Power Grid network.

Considering that the goal of most methods for measuring the diffusion power of nodes to select influential nodes among the top nodes of the list for further applications such as viral marketing, controlling outbreaks, and publishing innovations. Therefore, the proposed method has been able to increase the accuracy of ranking nodes, specially the top nodes of the in the first step by electing high-quality attributes and in the next step with an optimal combination list.

**4.2.2. Method separability value in ranking nodes.** Distinct rank allocation is another criterion for comparing node diffusion evaluation methods; in other words, for ranking methods, it is preferred if fewer nodes are assigned to each rank. Therefore, ideal methods that allocate every rank to a single node are ideal for this criterion. Tests use the monotonicity parameter (M) [43] to analyze different methods' node ranking distinguishability and separability.

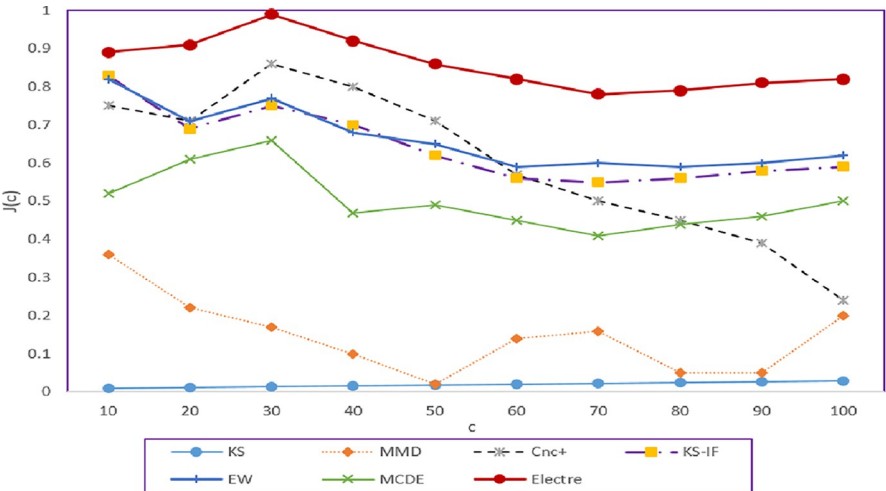

a.  NetScience

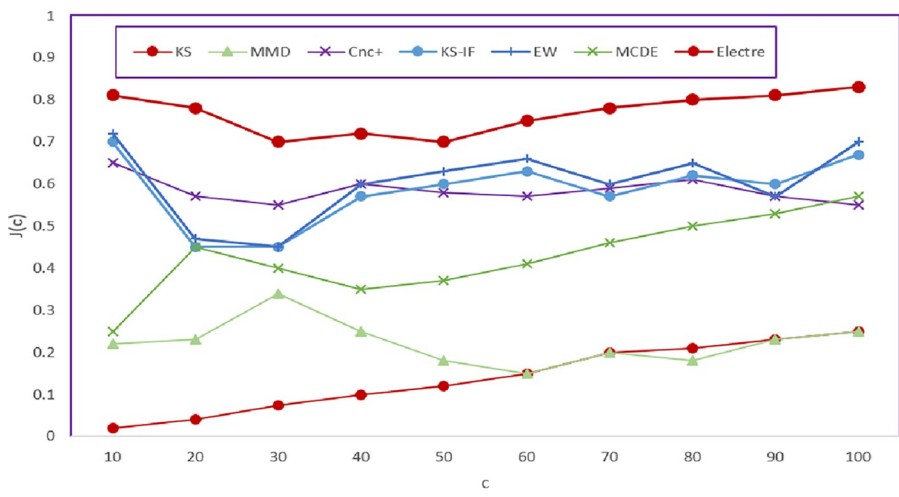

b.  PowerGrid

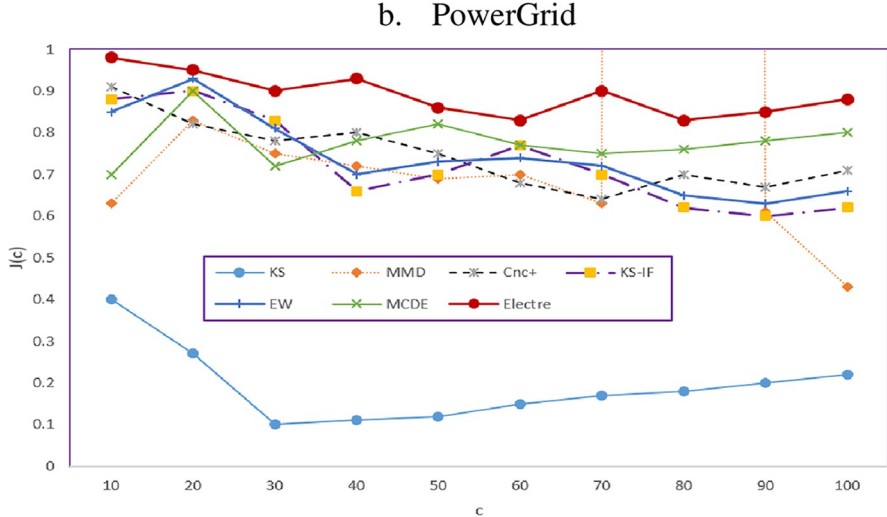

c. Elegans

**Fig 6. Accuracy of the proposed methods in assigning the top c ranks compared to different methods.** a. NetScience; b. PowerGrid; c. Elegans.

The monotonicity (M) of each ranking method executed on various datasets is shown in Table 5.

The results from Table 5 depict the proposed method's proper performance in most datasets; The quality of the method in allocating distinct ranks to nodes increases due to the method performance into the attention to the different nodes based on their local position and neighboring structure lake of attention to these features makes other methods accuracy decreased considering the same nodes in the same ranks. The proposed method had similar or slightly different separability values in multiple datasets with the EW and MCDE methods.

The next test has been performed to determine whether the proposed methods are time-efficient. Fig 7 illustrates the average 100 execution times for different methods across different networks. Based on the results of this experiment, the proposed method has an acceptable time efficiency by changing the size of networks, despite using a combination of different indices. The main reason for the appropriate execution time of the proposed method is due to the selection of local and semi-local indicators with linear time complexity.

## 5. Conclusion

This paper presented a method based on the simplified ÉLECTRE method to compare and rank nodes based on various indices. Index calculation time and accuracy were considered in selecting the effective structural indices to compare nodes. Therefore, the selected indices had a linear calculation time, and it was possible to extract them in large-scale networks with adequate speed. Regarding the high correlation of some indices with each other and their lower accuracy, a subset of the extracted indices was selected for the proposed method, and Shannon's entropy was used to determine the weight of each index. Results obtained based on various parameters indicated that the proposed method assigned distinct rankings to the nodes, such that it rarely occurred for two nodes to be ranked the same. Also, by increasing the infection rate of nodes, it was observed that the proposed method achieved better performance in ranking nodes. In addition, the method also performed very efficiently in ranking highly influential nodes. Given the power law distribution of node degrees in complex networks, the computation speed for the proposed method can be remarkably increased by removing nodes with

**Table 5. Monotonicity of methods in assigning distinct ranks to nodes.**

| Dataset | M(KS) | M(MMD) | M($C_{nc+}$) | M(KS-IF) | M(EW) | M(MCDE) | M(Electre) |
|---|---|---|---|---|---|---|---|
| Zebra | 0.3478 | 0.4219 | 0.8786 | 0.8786 | 0.8786 | 0.8901 | **0.9109** |
| Karate | 0.4958 | 0.7536 | 0.9472 | **0.9542** | **0.9542** | **0.9542** | **0.9542** |
| Contiguous | 0.1666 | 0.8171 | 0.9848 | 0.9949 | **1** | 0.9922 | 0.9966 |
| Dolphins | 0.3769 | 0.9041 | 0.9873 | **0.9979** | **0.9979** | **0.9979** | **0.9979** |
| Copperfield | 0.5990 | 0.9181 | 0.9968 | 0.9977 | **0.9997** | **0.9997** | **0.9997** |
| Netsciense | 0.6421 | 0.8215 | 0.9893 | 0.9946 | **0.9950** | 0.9945 | **0.9950** |
| Elegans | 0.8413 | 0.9277 | 0.9984 | 0.9980 | 0.9986 | 0.9985 | **0.9989** |
| Email | 0.8088 | 0.9229 | 0.9991 | 0.9996 | **0.9999** | **0.9999** | **0.9999** |
| Euroroad | 0.2126 | 0.6498 | 0.9175 | 0.9618 | **0.9863** | 0.9833 | 0.9573 |
| Hamsterster | 0.8714 | 0.9264 | 0.9855 | 0.9855 | 0.9853 | 0.9853 | **0.9899** |
| PowerGrid | 0.2460 | 0.6928 | 0.9420 | 0.9806 | **0.9970** | 0.9902 | 0.9811 |
| PGP | 0.4806 | 0.6678 | 0.9851 | 0.9906 | **0.9990** | **0.9990** | **0.9994** |

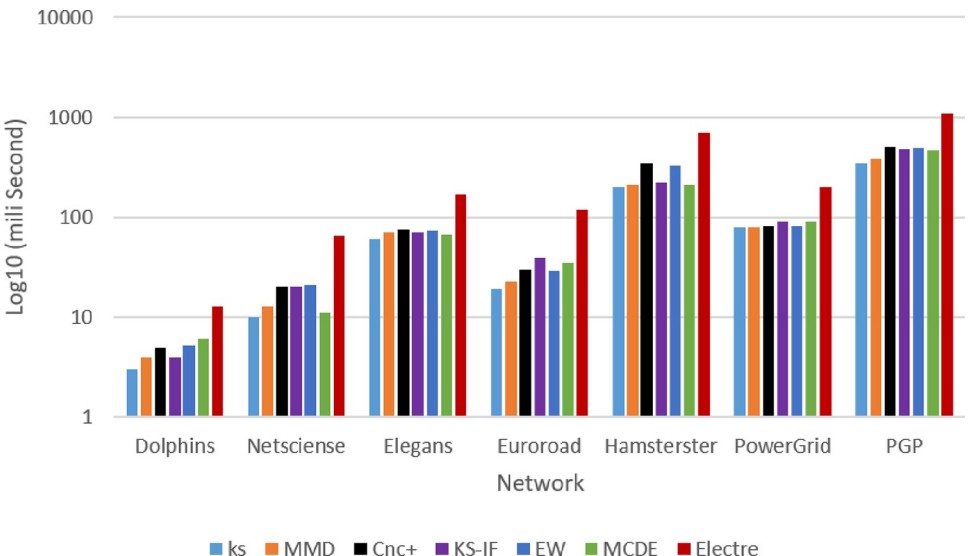

**Fig 7. Average execution time of the proposed method in comparison with other methods in different networks.**

a lower degree that generally have low diffusion power. This paper only uses structural features extracted from unweighted and directionless networks to present a multi-index method. To use it in weighted and directed networks, features related to the centrality index in these networks can be extracted and utilized.

## Author Contributions

**Conceptualization:** Amir Sheikhahmadi.

**Data curation:** Adib Sheikhahmadi, Shahnaz Mohammadimajd.

**Formal analysis:** Adib Sheikhahmadi, Shahnaz Mohammadimajd.

**Methodology:** Adib Sheikhahmadi, Amir Sheikhahmadi.

**Resources:** Farshid Veisi, Amir Sheikhahmadi, Shahnaz Mohammadimajd.

**Software:** Adib Sheikhahmadi, Farshid Veisi.

**Supervision:** Amir Sheikhahmadi.

**Validation:** Amir Sheikhahmadi, Shahnaz Mohammadimajd.

**Visualization:** Adib Sheikhahmadi, Farshid Veisi.

**Writing – original draft:** Adib Sheikhahmadi, Amir Sheikhahmadi.

**Writing – review & editing:** Farshid Veisi, Amir Sheikhahmadi.

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
