## [Decision Letter · Decision Letter 0]

10 Jul 2022

PONE-D-22-15682A Multi-Attribute Method for Ranking Influential nodes in Complex NetworksPLOS ONE

Dear Dr. Sheikhahmadi,

Thank you for submitting your manuscript to PLOS ONE. After careful consideration, we feel that it has merit but does not fully meet PLOS ONE’s publication criteria as it currently stands. Therefore, we invite you to submit a revised version of the manuscript that addresses the points raised during the review process.

We look forward to receiving your revised manuscript.

Kind regards,

Ali Safaa Sadiq

Academic Editor

PLOS ONE

Journal Requirements:

A clean copy of the edited manuscript (uploaded as the new *manuscript* file).

3. Please ensure you provide full context for the ELECTRE method used in this study, including discussion of any previous work that has used this method, and appropriate literature references.

NO authors have competing interests

The authors have declared that no competing interests exist.

5. PLOS requires an ORCID iD for the corresponding author in Editorial Manager on papers submitted after December 6th, 2016. Please ensure that you have an ORCID iD and that it is validated in Editorial Manager. To do this, go to ‘Update my Information’ (in the upper left-hand corner of the main menu), and click on the Fetch/Validate link next to the ORCID field. This will take you to the ORCID site and allow you to create a new iD or authenticate a pre-existing iD in Editorial Manager. Please see the following video for instructions on linking an ORCID iD to your Editorial Manager account: https://www.youtube.com/watch?v=_xcclfuvtxQ.

Reviewers' comments:

Reviewer's Responses to Questions

**Comments to the Author**

1. Is the manuscript technically sound, and do the data support the conclusions?

Reviewer #1: Partly

Reviewer #2: Yes

Reviewer #3: Partly

2. Has the statistical analysis been performed appropriately and rigorously? 

Reviewer #1: I Don't Know

Reviewer #2: I Don't Know

Reviewer #3: Yes

3. Have the authors made all data underlying the findings in their manuscript fully available?

Reviewer #1: Yes

Reviewer #2: Yes

Reviewer #3: No

4. Is the manuscript presented in an intelligible fashion and written in standard English?

Reviewer #1: Yes

Reviewer #2: No

Reviewer #3: No

5. Review Comments to the Author

Reviewer #1: The contribution of this paper is good and I am happy to endorse its acceptance at some point. However, there are several major and minor comments to address. I have listed them as follows:

• First off, please clearly state the gap targeted in this paper at the end of introduction and list down the hypotheses

• In terms of research method and design, there is not much in the paper.

• The comparative algorithms in the experiments are not properly acknowledged and cited

• I also suggest adding some figures to better articular the content as the paper looks very dry at the moment.

• Analysis of the results is missing in the paper. There is a big gap between the results and conclusion. There should be the result analysis between these two sections. After comparing the numerical methods, you have to be able to analyse the results and relate them to their structures. It would be interesting to have your thoughts on why the method works that way? Such analyses would be the core of your work where you prove your understanding of the reason behind the results. You can also link the findings to the hypotheses of the paper. Long story short, this paper requires a very deep analysis from different perspectives

• There is no statistical test to judge about the significance of the numerical method’s results. Without such a statistical test, the conclusion cannot be supported

• There is no discussion on the cost effectiveness of the proposed method. What is the computational complexity? What is the runtime? Please include such discussions. You can also use the big oh notation to show the computation complexity.

• Some mathematical notations and Lemma presentations are not rigorous enough to correctly understand the contents of the paper. The authors are requested to recheck all the definition of variables and further clarify these equations.

Reviewer #2: In this paper the authors propose a method to determine and rank node diffusion powers using various attributes.

In my opinion, the paper is interesting but it is not well structured and there are some weaknesses that must be solved. The first one is that I am not sure if its the contribution to the research field is relevant enough.

ABSTRACT

The abstract must be rewritten because of it do not is a summary of the paper. It is a copy of a part of the introduction.

INTRODUCTION

Regarding to section 1 (Introduction), It does not explain the motivation that leads the authors to carry out the paper.

Why is the article written? What does it mean that they don't have other articles already written?

In the introduction, they insists that the accuracy is increased, based on what?

I think you can't say that without saying why.

Also, when they comment on the three elements of innovation of the study, suddenly ELECTRE appears without reference or anything, what is that?

LITERATURE REVIEW

I don't understand this section, why are there two subsections?

Why are seven measures chosen? Aren't there more?

Are these seven metrics going to be used to make a comparison?

Nothing is said.

In 2.2 subsection the paper says that there are two methods to analyze the diffusion power of nodes in social networks. All the time are talking about diffusion power in complex networks and suddenly they are limited to social networks, I don't understand. In addition, they say that one method is to calculate the node diffusion power in a real manner, I am not clear that this is a method, it is a kind of brute force to analyze the network and it is not used.

There are many different diffusion evaluation methods: the cascade method, the linear threshold method, the SIR method, etc. Why do they use the SIR model? Why is it the most famous?

THE PROPOSE METHOD

In the propose method section a flowchart appears, which is very important, but there are elements that need to be changed or better explained. Again the ELECTRE method appears without explanation. What does it consist of? Is it a James Bond method :)) ?

Many methods use network structures to calculate the node powers. Why the authors use these ten methods? Also, some of them have not been explained in Literature review, there is a contradiction.

Another important thing is that they suddenly remove Betweenness and Closeness basically due to their complexity on large networks. But it is obvious that there are methods for calculating Betweenness with lower computational cost, even without calculating the shortest path. It seems that they are eliminated by superficial criteria. Also, if they are not going to be chosen, it would not have been better not to put them in section 3.2 and the problem is over.

The authors base the selection on the correlation, using the Spearman coefficient, between the centrality measures and the result of the SIR model. I'm sure if other flow based measures are used the correlation would be higher, which leads me to think that the result of the method they present depends on the methods used.

EXPERIMENTAL ANALYSIS

This part seems to be the best part of the paper and I don't think many changes would be needed. In fact, until I have reached this part I have understood almost nothing.

In short, I believe that the paper is novel and has the potential to be published, but it suffers from a lack of clarity, especially in the first sections.

Reviewer #3: This paper has a potential to be accepted, but some important points have to be clarified or fixed before we can proceed and a positive action can be taken.

- The manuscript needs extensive English editing because there are several typos and grammatical errors.

- It is really unclear to me why the authors applied SIR diffusion model. What is the specific property of this model?

- The literature review section is very poor. There are several descent research efforts that can be considered in this section.

- The legends in Fig. 5 are missing.

- I'm not convinced that high correlation of an attribute with the diffusion power in SIR model could be a good reason to be selected for ranking. Can other dimensionality reduction methods such as PCA give more precise results?

- What do the authors mean by "making an attribute scale-free" in equation (2)?

- The authors failed to explain why the ELECTRE method is chosen. The reader needs some more motivation.

6. PLOS authors have the option to publish the peer review history of their article (what does this mean?). If published, this will include your full peer review and any attached files.

Reviewer #1: No

Reviewer #2: No

Reviewer #3: No

---

## [Author Response · Author response to Decision Letter 0]

7 Sep 2022

Manuscript Ref. No.: PONE-D-22-15682

Dear Editor,

Reply to reviewers on PONE-D-22-15682

We would like first to thank the editor and the reviewers for helpful, constructive comments and observations that significantly contributed to improving the manuscript. Below we have concisely addressed each of the comments and suggestions and those changes implemented in the manuscript.

Also, we would like to note that the changes made in the text have been highlighted in blue in the new version of the manuscript to allow the reviewers to identify the newly added text easily.

Below you can find detailed explanations regarding each particular comment and the applied revision in the revised version of the manuscript.

Yours Sincerely,

Amir Sheikhahmadi, PhD 

Assistant Professor, Islamic Azad University, 

Sanandaj branch, Sanandaj, Iran 

Email: sheikhahmadi@eng.ui.ac.ir

 

Reviewers' comments:

Reviewer #1: 

The contribution of this paper is good and I am happy to endorse its acceptance at some point. However, there are several major and minor comments to address. 

Authors reply: We thank the reviewer for taking the time to read the manuscript and providing positive comments. We have tried our best to reply to your comments below.

1. First off, please clearly state the gap targeted in this paper at the end of introduction and list down the hypotheses

 Authors Reply: Thanks for your valuable comments. Items have been added to the introduction section.

2. In terms of research method and design, there is not much in the paper.

Authors Reply: proper changes were done in the introduction and related works sections. 

3. The comparative algorithms in the experiments are not properly acknowledged and cited.

Authors reply: Thanks for your consideration. All of the methods were added and referenced in the literature review section. 

4. I also suggest adding some figures to better articular the content as the paper looks very dry at the moment.

Authors reply: I agree entirely. For better understanding, several figures were added in different sections.

5. Analysis of the results is missing in the paper. There is a big gap between the results and conclusion. There should be the result analysis between these two sections. After comparing the numerical methods, you have to be able to analyse the results and relate them to their structures. It would be interesting to have your thoughts on why the method works that way? Such analyses would be the core of your work where you prove your understanding of the reason behind the results. You can also link the findings to the hypotheses of the paper. Long story short, this paper requires a very deep analysis from different perspectives.

There is no statistical test to judge about the significance of the numerical method’s results. Without such a statistical test, the conclusion cannot be supported.

Authors reply: 

Thanks for your careful reading. Two new sections have been added to the evaluations for comparison and analysis. A portion of the tests is dedicated to testing the validity of the top-ranked nodes on the ranked list because those nodes are more important than those at the bottom. To accomplish this, the similarity c of the initial element of the list R ranked by each method with the ranking list of the actual publication σ is calculated.

6. There is no discussion on the cost effectiveness of the proposed method. What is the computational complexity? What is the runtime? Please include such discussions. You can also use the big oh notation to show the computation complexity.

Authors reply: In the evaluation section, a comparison of the execution time of the proposed method compared to other added methods.

7. Some mathematical notations and Lemma presentations are not rigorous enough to correctly understand the contents of the paper. The Authorss are requested to recheck all the definition of variables and further clarify these equations.

Authors reply: Thank you, all the items were reviewed and corrected

Reviewer #2:

In this paper the Authorss propose a method to determine and rank node diffusion powers using various attributes. In my opinion, the paper is interesting but it is not well structured and there are some weaknesses that must be solved.

Authors reply: Thanks for your valuable comments. We have tried our best to reply to your comments below.

1. The abstract must be rewritten because of it do not is a summary of the paper. It is a copy of a part of the introduction.

Authors reply: Thanks for your consideration. The abstract was modified.

2. Regarding to section 1 (Introduction), It does not explain the motivation that leads the Authorss to carry out the paper. Why is the article written? What does it mean that they don't have other articles already written? In the introduction, they insist that the accuracy is increased, based on what? I think you can't say that without saying why. Also, when they comment on the three elements of innovation of the study, suddenly ELECTRE appears without reference or anything, what is that?

Authors reply: Thanks, the introduction section was modified and the requested items added.

3. I don't understand this section, why are there two subsections? Why are seven measures chosen? Aren't there more? Are these seven metrics going to be used to make a comparison? In 2.2 subsection the paper says that there are two methods to analyze the diffusion power of nodes in social networks. All the time are talking about diffusion power in complex networks and suddenly they are limited to social networks, I don't understand. In addition, they say that one method is to calculate the node diffusion power in a real manner, I am not clear that this is a method, it is a kind of brute force to analyze the network and it is not used.

There are many different diffusion evaluation methods: the cascade method, the linear threshold method, the SIR method, etc. Why do they use the SIR model? Why is it the most famous?

Authors reply: Regarding the use of the SIR model, it should be noted that this model is a generalized IC model, where the probability of infected nodes improving at the end of each stage is 1. However, in the linear threshold model, a threshold is needed to activate each node, and because this threshold is generally unavailable, the SIR model has been used in many articles and utilized in our study.

4. In the propose method section a flowchart appears, which is very important, but there are elements that need to be changed or better explained. Again the ELECTRE method appears without explanation. What does it consist of? Is it a James Bond method :)) ?

Many methods use network structures to calculate the node powers. Why the Authorss use these ten methods? Also, some of them have not been explained in Literature review, there is a contradiction.

Another important thing is that they suddenly remove Betweenness and Closeness basically due to their complexity on large networks. But it is obvious that there are methods for calculating Betweenness with lower computational cost, even without calculating the shortest path. It seems that they are eliminated by superficial criteria. Also, if they are not going to be chosen, it would not have been better not to put them in section 3.2 and the problem is over.

The Authorss base the selection on the correlation, using the Spearman coefficient, between the centrality measures and the result of the SIR model. I'm sure if other flow based measures are used the correlation would be higher, which leads me to think that the result of the method they present depends on the methods used.

Authors reply: Your consideration and excellent vision are respectable. The proposed method was altered based on your valuable comments. Closeness and Betweenness attributes were removed.

5. This part seems to be the best part of the paper and I don't think many changes would be needed. In fact, until I have reached this part I have understood almost nothing. In short, I believe that the paper is novel and has the potential to be published, but it suffers from a lack of clarity, especially in the first sections.

Authors reply: Thanks for your valuable comments. Initial sections were changed and two new comparisons were added to the evaluation section as well.

Reviewer #3:

This paper has a potential to be accepted, but some important points have to be clarified or fixed before we can proceed and a positive action can be taken.

Authors reply: Thanks for valuable comments. We have tried our best to reply your comments in below.

1. The manuscript needs extensive English editing because there are several typos and grammatical errors.

Authors reply: Thanks.All sections have been revised.

2. It is really unclear to me why the Authorss applied SIR diffusion model. What is the specific property of this model?

Authors reply: This is an excellent suggestion to be used in future works. In this study, the implementation of the SIR model with many iterations to calculate the propagation power of nodes can indicate the reality of propagation in networks, . this model has been used in many articles. 

3. The literature review section is very poor. There are several descent research efforts that can be considered in this section.

Authors reply: The literature reviews completely modified and new references, added to this section. 

4.The legends in Fig. 5 are missing.

Authors reply: Thanks, It is modified .

5.I'm not convinced that high correlation of an attribute with the diffusion power in SIR model could be a good reason to be selected for ranking. Can other dimensionality reduction methods such as PCA give more precise results?

Authors reply: This is an excellent suggestion to be used in future works. In this study, the implementation of the SIR model with many iterations to calculate the propagation power of nodes can indicate the reality of propagation in networks, this model has been used in many articles. 

6.What do the Authorss mean by "making an attribute scale-free" in equation (2)?

Authors reply: Using an attribute scale-free made these features comparable, considering, different measurement scale of various attributes, so that the attributes can be measured without dimensions. 

7. The Authors failed to explain why the ELECTRE method is chosen. The reader needs some more motivation.

Authors reply: I appreciate your consideration, in the Abstract and the proposed method section essential explanations, were added.

---

## [Decision Letter · Decision Letter 1]

22 Sep 2022

PONE-D-22-15682R1A Multi-Attribute Method for Ranking Influential nodes in Complex NetworksPLOS ONE

Dear Dr. Sheikhahmadi,

Thank you for submitting your manuscript to PLOS ONE. After careful consideration, we feel that it has merit but does not fully meet PLOS ONE’s publication criteria as it currently stands. Therefore, we invite you to submit a revised version of the manuscript that addresses the points raised during the review process.

We look forward to receiving your revised manuscript.

Kind regards,

Ali Safaa Sadiq

Academic Editor

PLOS ONE

Journal Requirements:

Additional Editor Comments :

Authors are invited to address the given minor comments by reviewers 1 and 2 and provide their revised version along with the detailed response letter.

Reviewers' comments:

Reviewer's Responses to Questions

**Comments to the Author**

1. If the authors have adequately addressed your comments raised in a previous round of review and you feel that this manuscript is now acceptable for publication, you may indicate that here to bypass the “Comments to the Author” section, enter your conflict of interest statement in the “Confidential to Editor” section, and submit your "Accept" recommendation.

Reviewer #1: (No Response)

Reviewer #2: All comments have been addressed

Reviewer #3: All comments have been addressed

2. Is the manuscript technically sound, and do the data support the conclusions?

Reviewer #1: (No Response)

Reviewer #2: Yes

Reviewer #3: Yes

3. Has the statistical analysis been performed appropriately and rigorously? 

Reviewer #1: (No Response)

Reviewer #2: Yes

Reviewer #3: Yes

4. Have the authors made all data underlying the findings in their manuscript fully available?

Reviewer #1: (No Response)

Reviewer #2: Yes

Reviewer #3: Yes

5. Is the manuscript presented in an intelligible fashion and written in standard English?

Reviewer #1: (No Response)

Reviewer #2: No

Reviewer #3: Yes

6. Review Comments to the Author

Reviewer #1: Some final cosmetic comments:

* The results of your comparative study should be discussed in-depth and with more insightful comments on the behaviour of your algorithm on various case studies. Discussing results should not mean reading out the tables and figures once again.

* Avoid lumping references as in [x, y] and all other. Instead summarize the main contribution of each referenced paper in a separate sentence. For scientific and research papers, it is not necessary to give several references that say exactly the same. Anyway, that would be strange, since then what is innovative scientific contribution of referenced papers? For each thesis state only one reference.

* Avoid using first person.

* Avoid using abbreviations and acronyms in title, abstract, headings and highlights.

* Please avoid having heading after heading with nothing in between, either merge your headings or provide a small paragraph in between.

* The first time you use an acronym in the text, please write the full name and the acronym in parenthesis. Do not use acronyms in the title, abstract, chapter headings and highlights.

* The results should be further elaborated to show how they could be used for the real applications.

Reviewer #2: Almost all of the corrections suggested in the first round have been ironed out however, the manuscript needs extensive English editing because of there are typos and grammatical errors.

Reviewer #3: I have no more comments.

7. PLOS authors have the option to publish the peer review history of their article (what does this mean?). If published, this will include your full peer review and any attached files.

Reviewer #1: No

Reviewer #2: No

Reviewer #3: No

---

## [Author Response · Author response to Decision Letter 1]

17 Oct 2022

Manuscript Ref. No.: PONE-D-22-15682

Dear Editor,

Reply to reviewers on PONE-D-22-15682

We would like first to thank the editor and the reviewers for helpful, constructive comments and observations that significantly contributed to improving the manuscript. Below we have concisely addressed each of the comments and suggestions and those changes implemented in the manuscript.

Also, we would like to note that the changes made in the text have been highlighted in blue in the new version of the manuscript to allow the reviewers to identify the newly added text easily.

Below you can find detailed explanations regarding each particular comment and the applied revision in the revised version of the manuscript.

Yours Sincerely,

Amir Sheikhahmadi, PhD 

Assistant Professor, Islamic Azad University, 

Sanandaj branch, Sanandaj, Iran 

Email: sheikhahmadi@eng.ui.ac.ir

 

Reviewers' comments:

Reviewer #1: 

1. The results of your comparative study should be discussed in-depth and with more insightful comments on the behavior of your algorithm on various case studies. Discussing results should not mean reading out the tables and figures once again.

Authors reply: Thanks for your valuable comment. In the revised version, All results were analyzed and added to the text. 

2. Avoid lumping references as in [x, y] and all other. Instead summarize the main contribution of each referenced paper in a separate sentence. For scientific and research papers, it is not necessary to give several references that say exactly the same. Anyway, that would be strange, since then what is innovative scientific contribution of referenced papers? For each thesis state only one reference.

Authors Reply: proper changes were done in the whole part of the manuscript.

¬

3. Avoid using first person.

Authors reply: Thanks for your consideration. It is done.

4. Avoid using abbreviations and acronyms in title, abstract, headings and highlights.

Authors reply: the proper modification is done .

5. Please avoid having heading after heading with nothing in between, either merge your headings or provide a small paragraph in between.

Authors reply: It is done.

6. There is The first time you use an acronym in the text, please write the full name and the acronym in parenthesis. Do not use acronyms in the title, abstract, chapter headings and highlights.

Authors reply: Thanks, It is done.

7. The results should be further elaborated to show how they could be used for the real applications.

Authors reply: Thank you, all the items were reviewed and corrected

Reviewer #2:

Almost all of the corrections suggested in the first round have been ironed out however, the manuscript needs extensive English editing because of there are typos and grammatical errors.

Authors reply: Thanks for your valuable comments which help us to improve manuscript. All sections have been revised carefully.

Reviewer #3:

I have no more comments.

Authors reply: Thanks for your valuable comments which help us to improve manuscript.

---

## [Decision Letter · Decision Letter 2]

10 Nov 2022

A Multi-Attribute Method for Ranking Influential nodes in Complex Networks

PONE-D-22-15682R2

Dear Dr. Sheikhahmadi,

We’re pleased to inform you that your manuscript has been judged scientifically suitable for publication and will be formally accepted for publication once it meets all outstanding technical requirements.

Kind regards,

Ali Safaa Sadiq

Academic Editor

PLOS ONE

Additional Editor Comments (optional):

The authors have addressed all the given comments by reviewers, hence I am happy to recommend their paper for the possible publication.

Reviewers' comments:

Reviewer's Responses to Questions

**Comments to the Author**

1. If the authors have adequately addressed your comments raised in a previous round of review and you feel that this manuscript is now acceptable for publication, you may indicate that here to bypass the “Comments to the Author” section, enter your conflict of interest statement in the “Confidential to Editor” section, and submit your "Accept" recommendation.

Reviewer #1: (No Response)

Reviewer #2: All comments have been addressed

2. Is the manuscript technically sound, and do the data support the conclusions?

Reviewer #1: (No Response)

Reviewer #2: Yes

3. Has the statistical analysis been performed appropriately and rigorously? 

Reviewer #1: (No Response)

Reviewer #2: Yes

4. Have the authors made all data underlying the findings in their manuscript fully available?

Reviewer #1: (No Response)

Reviewer #2: Yes

5. Is the manuscript presented in an intelligible fashion and written in standard English?

Reviewer #1: (No Response)

Reviewer #2: Yes

6. Review Comments to the Author

Reviewer #1: all comments have been addressed. all comments have been addressed. all comments have been addressed.

Reviewer #2: (No Response)

7. PLOS authors have the option to publish the peer review history of their article (what does this mean?). If published, this will include your full peer review and any attached files.

Reviewer #1: No

Reviewer #2: No

---

## [Editor Report · Acceptance letter]

14 Nov 2022

PONE-D-22-15682R2 

A Multi-Attribute Method for Ranking Influential nodes in Complex Networks 

Dear Dr. Sheikhahmadi:

I'm pleased to inform you that your manuscript has been deemed suitable for publication in PLOS ONE. Congratulations! Your manuscript is now with our production department. 

Kind regards, 

on behalf of

Dr. Ali Safaa Sadiq 

Academic Editor

PLOS ONE